# Bridging the Gap: Aligning Language Model Generation with Structured Information Extraction via Controllable State Transition

Submission Id: 1168

## Abstract

Large language models (LLM) achieve superior performance in generative tasks. However, due to the natural gap between language model generation and structured information extraction in three dimensions: task type, output format, and modeling granularity, they often fall short in structured information extraction, a crucial capability for effective data utilization on the web. In this paper, we define the generation process of the language model as the controllable state transition, aligning the generation and extraction processes to ensure the integrity of the output structure and adapt to the goals of the information extraction task. Furthermore, we propose the Structure2Text decider to help the language model understand the fine-grained extraction information, which converts the structured output into natural language and makes state decisions, thereby focusing on the task-specific information kernels, and alleviating language model hallucinations and incorrect content generation. We conduct extensive experiments and detailed analyses on myriad information extraction tasks. Our method not only achieves significant performance improvements but also ensures the integrity of the output structure, making it easy to parse the extracted content.

## CCS Concepts

• **Computing methodologies** → **Information extraction**.

## Keywords

Information Extraction, Large Language Model, Few-shot Learning, Structure Generation

**ACM Reference Format:**
Anonymous Author(s). 2018. Bridging the Gap: Aligning Language Model Generation with Structured Information Extraction via Controllable State Transition. In *Proceedings of Make sure to enter the correct conference title from your rights confirmation emai (Conference acronym 'XX).* ACM, New York, NY, USA, 10 pages. https://doi.org/XXXXXXX.XXXXXXX

## 1 Introduction

Large language models have gained widespread popularity due to their superior performance in generative tasks [1, 14], producing

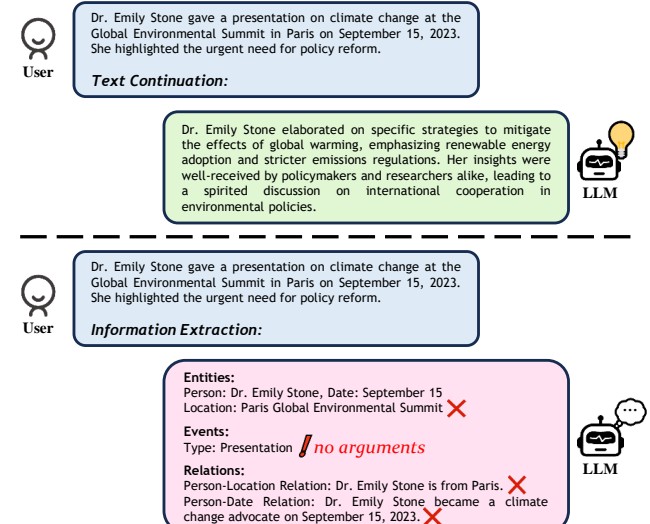

**Figure 1: The gap between language model generation and structured information extraction.**

contextually rich and coherent content that enhances user interactions across diverse web platforms [51]. However, despite these advances, they often fall short in structured information extraction, a crucial capability for effective data utilization on the web [2]. This deficiency is particularly evident in web environments where accurate information extraction and organization are essential for applications such as semantic search [8, 16], content recommendation systems [35], and automated knowledge base updates [27].

It is intuitive that such discrepancies arise: the fundamental objectives of language model generation and structured information extraction inherently diverge. Language models, trained on extensive textual datasets, are designed to predict the next token or generate coherent, semantically rich text. So language models primarily focus on linguistic fluency, semantic coherence, and understanding context. Conversely, structured information extraction is tasked with pulling specific, meaningful information from unstructured text and organizing it into structured forms like relational triples or event tuples, emphasizing the accuracy, completeness, and adherence to predefined structures of the data. Therefore, a natural gap exists between language model generation and structured information extraction.

To further understand the inherent reasons behind this gap, a comparison between language model generation and structured information extraction can be elucidated across three dimensions: task type, output format, and modeling granularity.

- **Generation vs. Extraction**. The core task of language models is to generate natural language, which may produce redundant or unnecessary information. In contrast, structured information extraction is focused on extracting and organizing precise information, demanding accuracy.
- **Freedom vs. Structure**. Language models enjoy considerable freedom in text generation, producing content that is open-ended and unstructured. However, information extraction requires outputs to be highly structured, adhering to predefined rules or formats.
- **Coarser-grained vs. Fine-grained**. Language models excel in understanding context and focus on coarser-grained information, whereas information extraction typically demands attention to fine-grained details within the text, mapping them accurately to predefined categories.

*How to bridge the gap and make language model generation efficient for structured information extraction?* To this end, we define the language model generation process as the Controllable **S**tate **T**ransition and incorporate the goals of the information extraction task into the state decision process, to align language model **G**eneration and information **E**xtraction (STGE). Specifically, we define five states based on the features of the information extraction task, which can efficiently represent the generation of labels and corresponding extracted content. Furthermore, we propose the Structure2Text decider to help the language model understand the fine-grained extraction information, which converts the structured output into natural language and makes state decisions, thereby focusing on the task-specific information kernels, and alleviating language model hallucinations [21] and incorrect content generation. We conduct extensive experiments and detailed analyses on named entity recognition, relation extraction, and event argument extraction tasks. Our method not only achieves significant performance improvements but also ensures the integrity of the output structure, making it easy to parse the extracted content.

The contributions are summarized as follows:

- We define the generation process of the language model as the controllable state transition, aligning the generation and extraction processes to ensure the integrity of the output structure and adapt to the goals of the information extraction task.
- We propose Structure2Text decider to convert output structure into natural language and make state decisions to focus on fine-grained information in text, alleviating language model hallucinations and incorrect content generation.
- We conduct extensive experiments and detailed analysis on myriad information extraction tasks, demonstrating that our method achieves significant performance improvements in multiple scenarios.

## 2 Related Work

### 2.1 Generative Information Extraction

Benefiting from the excellent contextual reasoning capabilities of the large language model, some works have proposed integrating the large language model into information extraction tasks based on the generation paradigm [47]. Generation-based methods require designing the output format and parsing the extracted content and corresponding labels from the output.

*Name Entity Recognition (NER).* Researchers [7, 44] propose NER methods based on In-context Learning [4]. GPT-NER [44] uses special tokens to mark the entities that need to be extracted in the output, and proposes a self-verification strategy to alleviate the hallucination issue of the language model and over-confidently label NULL inputs as entities. MetaNER [7] design output is "ENTITY is type", and injects in-context NER ability into the pre-trained language model, which can recognize new types of entities using only a few demonstrations. GNER [11] designs the output in the format of sequence tagging, identifies the entity type of each word one by one, and explores the impact of negative instances on NER. Researchers [3] analyze different context demonstration selection methods for NER in scientific documents.

*Relation Extraction (RE).* QA4RE [49] obtains the corresponding entity relationship through question and answer based on specific relation templates. SUMASK [25] recursively uses the large language model to transform RE input into the effective question-answering format. ERA-CoT [32] captures the relationships between entities to help the large language model understand the context, and improves reasoning ability through Chain-of-Thoughts (CoT) [46]. GPT-RE [43] designs specific prompts and reasoning logic process, and queries the language model about the relation between entities to complete the RE task. MICRE [26] designs output as the tabular format using "|" as the recognizable delimiter of tables, and it learns new RE tasks in context more effectively through meta-training, thereby achieving better generalization on zero-shot and few-shot tasks. TableIE [24] defines RE as the table generation task and uses "|" as the delimiter, which incorporates explicit structured information into in-context learning, thus facilitating the conversion of output into RE structure.

*Event Argument Extraction (EAE).* Researchers [20, 28, 36, 45] designed specific prompts and output structures to complete the EAE task in a generative paradigm. Specifically, BART-Gen [28] and DEGREE [20] construct prompts for each event type, guiding the language model to generate corresponding arguments in the role slot. TEXT2EVENT [36] converts events into the tree structure, uses the depth-first algorithm to convert the tree structure into the linear sequence, and generates arguments in the text generation paradigm. CODE4STRUCT [45] designs the output structure as code, using the features of the programming language to complete the EAE task in the code generation manner. Researchers [13, 39] applied Retrieval-Augmented Generation to the EAE task to enhance the performance of the EAE task by retrieving demonstrations that are suitable for the current context.

### 2.2 Controllable Text Generation

Recently, large language models have demonstrated high quality in text generation and have attracted a large number of users on the web. However, in practical applications, large language models must meet increasingly complex user requirements, including semantic control such as toxicity [30], topic [5, 9], sentiment [6, 23], style [22], lexical control such as keyword or phrase inclusion [18, 50], and structural control such as tables [24], poetry [48, 52], recipes

**Generation Framework**

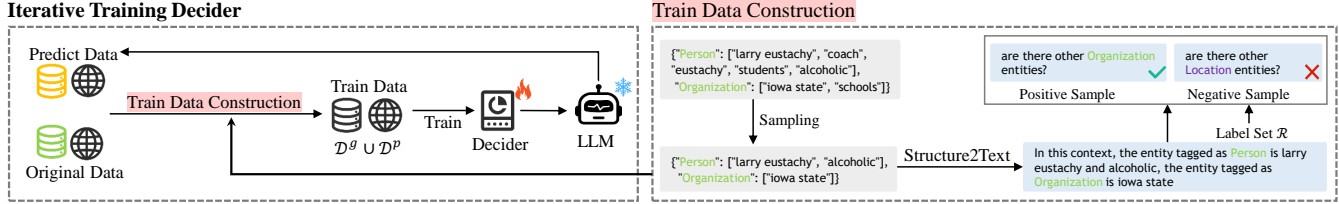

Figure 2: Overall framework of our method STGE. In the Generation Framework, taking the NER task as an example, we use an example in the ACE05 dataset to illustrate this process (Input only shows the context, the complete input can be found in Figure 3), and we show the decision process from $S^g_{value}$ to $S^f_{value}$ or $S^g_{value}$ as an example. In the Iterative Training Decider, we perform multiple rounds of iterative training on the Structure2Text Decider, constructing training data through the original data and the data predicted by the language model.

[33], etc. These diverse requirements have driven the rapid development of controllable text generation techniques, which ensure that the output meets predefined control conditions while maintaining high standards of helpfulness, fluency, and diversity [29]. Researchers [41] explore whether structured generation would limit the reasoning and domain knowledge understanding capabilities of large language models. StructuredRAG [40] proposes a benchmark designed to evaluate the ability of LLMs to follow response format instructions using different prompting strategies. Different from these methods, our method cleverly combines the features of large language models and information extraction through controllable state transitions, which not only ensures the correct generation format and reduces the complex parsing process, but also significantly improves the extraction ability of the model and enhances the performance of information extraction tasks.

## 3 Methodology

In this section, we delineate task formalization and the basic generation framework, followed by a detailed exposition of the controllable state transition mechanism. We further elaborate on the decision-making and training processes integral to the Structure2Text Decider. Figure 2 shows the overall framework.

### 3.1 Task Formalization

We first formalize the task of NER, RE, and EAE. Each task operates within a given a context $\mathbf{c} = \{c_1, ..., c_n\}$ consisting of $n$ words. For the NER task, given the set of entity types $\mathcal{R}^{ent}$, the NER task aims to extract all entities $\{e_1, ..., e_u\}$ in $\mathbf{c}$ and assign a type to each extracted entity $e_i$, where $e_i$ is the text span in the context $\mathbf{c}$. For the RE task, given the set of relation types $\mathcal{R}^{re}$, the RE task aims to extract all entity pairs $\{(e_o^1, e_s^1), ..., (e_o^u, e_s^u)\}$ in $\mathbf{c}$ and assign a type to each extracted entity pair $(e_o^i, e_s^i)$, where $e_o^i$ and $e_s^i$ are text spans in context $\mathbf{c}$. For the EAE task, given the event trigger word $c^{tri} \in \mathbf{c}$, the event type $t \in \mathcal{T}$ and the set of event-specific role types $\mathcal{R}^{event}_t$, the EAE task aims to extract all arguments $\{a_1, ..., a_u\}$ related to $c^{tri}$ in $\mathbf{c}$ and assign a role $r \in \mathcal{R}^{event}_t$ to each extracted argument $a_i$, where $a_i$ is a text span in the context $\mathbf{c}$.

### 3.2 Basic Generation Framework

This section outlines the basic generation framework tailored for information extraction tasks. Specifically, the input $\mathbf{x}$ to language model comprises several components: the task definition $\mathbf{z}$, the type set $\mathcal{R}$, the context $\mathbf{c}$, and additionally includes demonstration $\mathbf{m}_c$ in the few-shot scenario: $\mathbf{x} = [\mathbf{z}; \mathbf{m}_c; \mathcal{R}; \mathbf{c}]$, where $[;]$ denotes the

Task Definition **z**   Entity is words with specific meanings in the context, mainly including names of people, places, institutions, proper nouns, etc. Please output the entities in the following context. The format of the output is json, the key is the entity type, and the value is an entity list. Entity must be consecutive words in the context. If the entity type does not have a corresponding entity, the key and value will not be output.

Demonstration **m**$_c$   The format refers to the demonstration below:
                        Context: ......   Entity type: ......   Answer: ......

Type **R**   Entity type: Person, Organization, Geo-Political Entity, Vehicle, Location, Facility, Weapon

Context **c**   A week after these pictures surfaced, and after saying he 'd fight for his job, larry eustachy resigned as basketball coach at iowa state. the damage to his credibility was too much after eustachy admitted he partied with students at rival schools, and thathe's an alcoholic.

🤖
LLM

Output **y**   {"Person": ["larry eustachy", "coach", "eustachy", "students", "alcoholic"],
              "Organization": ["iowa state", "schools"]}

**Figure 3: A NER example from the ACE05 dataset, with details omitted due to space limitations.**

concatenation operation. The input of a NER example on the ACE05 dataset is shown in Figure 3.

In the few-shot scenario, demonstration selection is context-sensitive. We first use a language model to encode the current context $\mathbf{h}_c = \text{Encoder}(\mathbf{c})$ and few-shot data respectively. Then employing cosine similarity as the retrieval criterion to select the Top-k data entries $d_k$ from the few-shot data (such as 20-shot) as demonstrations $\mathbf{m}_c$ of context $\mathbf{c}$. Each demonstration $\mathbf{m}_c^i$ incorporates the context, the labels to be extracted (entity type, relation type, and argument role), and their corresponding extraction results.

$$\text{Sim}(\mathbf{h}_c^i|\mathbf{h}_c) = \frac{\text{Mean}(\mathbf{h}_c^i) \cdot \text{Mean}(\mathbf{h}_c)}{||\text{Mean}(\mathbf{h}_c^i)|| \times ||\text{Mean}(\mathbf{h}_c)||}, \quad (1)$$

$$d_k = \text{Top-k}_{d_i \in \mathcal{D}}(\text{Sim}(\mathbf{h}_c^i|\mathbf{h}_c)), \quad (2)$$

where Mean denotes the mean-pooling operation, $d_i \in \mathcal{D}$ refers to the data in the few-shot dataset, and $\mathbf{h}_c^i$ signifies the corresponding context representation.

During the generation process, the language model utilizes the previously generated tokens $\mathbf{y}_{<i}$ and input $\mathbf{x}$ to model the conditional probability of next token $\mathbf{y}_i$. Consequently, the total probability $p(\mathbf{y}|\mathbf{x})$ for generating output $\mathbf{y}$ is calculated as follows:

$$p(\mathbf{y}|\mathbf{x}) = \prod_i^{|\mathbf{y}|} p(\mathbf{y}_i|\mathbf{y}_{<i}, \mathbf{x}), \quad (3)$$

where output $\mathbf{y}$ adopts a JSON structure to represent the extraction result. As illustrated in Figure 3, each key in the output $\mathbf{y}$ corresponds to a label to be extracted, while the associated value is a list containing the corresponding extracted content, such as entities, entity pairs, and arguments.

### 3.3 Controllable State Transition

Compared with generation tasks, information extraction tasks possess two distinct features: **(1)** The extracted content must have specific labels. **(2)** The extracted content strictly originates from the initial context. Motivated by finite-state machine, we leverage these features alongside structured output to define the language model's decoding process as the controllable state transition, which includes the following five states:

- Start state $S_s$: Start state marks the beginning of the generation process, where the model outputs the initial character

**Table 1: The permissible subsequent states for each state.**

|  | $S_s$ | $S_{key}^g$ | $S_{value}^g$ | $S_{value}^f$ | $S_e$ |
|---|---|---|---|---|---|
| **Start** $S_s$ | × | ✓ | × | × | ✓ |
| **Generate Key** $S_{key}^g$ | × | × | ✓ | ✓ | × |
| **Generate Value** $S_{value}^g$ | × | × | ✓ | ✓ | × |
| **Finish Value** $S_{value}^f$ | × | ✓ | × | × | ✓ |
| **End** $S_e$ | - | | | | |

of the structured format, such as the opening brace '{' for JSON. This initiates the structured output, setting the stage for subsequent content generation.

- Generate Key state $S_{key}^g$: This state is dedicated to generating labels for the structured output, such as entity types, relation types, and argument roles. The label generated by this state must belong to the label set $\mathcal{R}$ and cannot be repeated with the label generated by the previous state.
- Generate Value state $S_{value}^g$: This state is tasked with generating the content of the corresponding label extracted from the context, which is the subsequent state of the Generate Key state. This state ensures that the generated content must come from the context.
- Finish Value state $S_{value}^f$: Finish Value State indicates the end of Generate Value State corresponding to the current Generate Key state, which generates the end character of the value state.
- End state $S_e$: End state indicates the end of the generation process. In this state, the model outputs the final character of the structured format, such as the closing brace '}' for JSON, finalizing the structured output.

The generation process of the language model can be conceptualized as a series of state transitions: $\mathbf{y} = [S_s, S_{key}^g, S_{value}^g, S_{value}^g, S_{value}^f, S_{key}^g, S_{value}^g, S_{value}^f, \ldots, S_e]$, where $\mathbf{y}$ means generating two keys and their corresponding values, the first key contains two corresponding values, and the second key contains one value.

To guarantee a complete and correct output structure, the transitions between states in the language model are carefully controlled, not arbitrary. The sequence begins with the Start State and concludes with the End State. Notably, the Generate Key State can be succeeded by multiple Generate Value States, indicating that the current label corresponds to multiple extracted contents. The permissible subsequent states for each state are detailed in Table 1.

### 3.4 Structure2Text Decider

State transition is a decision-making process where the model needs to predict the next state based on the current context and previous states:

$$P(S_{i+1}^j|\mathbf{c}, S_{<i}) = \frac{\text{Score}(S_{i+1}^j|\mathbf{c}, S_{<i})}{\sum_{S_{i+1}^k \in \mathcal{S}_{i+1}} \text{Score}(S_{i+1}^k|\mathbf{c}, S_{<i})}, \quad (4)$$

where $P(S_{i+1}^j|\mathbf{c}, S_{<i})$ represents the probability that the i+1 state is $S_{i+1}^j$, Score represents the scoring function, and $\mathcal{S}_{i+1}$ represents all possible subsequent states of the $S_i$ state in Table 1.

However, according to the modeling goal of the language model, the language model makes decisions at the token level and predicts the next token (Equation 3), so scoring function $\text{Score}(S_{i+1}^j | \mathbf{c}, S_{<i}) = p(\mathbf{y}|\mathbf{c}, S_{<i})[S_{i+1}^j]$ is token logits, where each state has specific preceding state and start characters, and we use the probability of the corresponding characters as the predicted probability of the state. As shown in the decision process example in Figure 2, the current state is $S_{value}^g$, the probability of character "]" represents the probability of the next state being $S_{value}^f$, and the probability of character "," represents the probability of the next state being $S_{value}^g$. This process focuses too much on linguistic fluency and ignores the connection between different states and the goal of information extraction. To this end, we propose the Structure2Text Decoder, which uses the information of previous states and the task-specific features to make state decisions in natural language.

Specifically, we first convert the previous state and label information into natural language, such as entities in NER, relations in RE, event type, trigger word and arguments in EAE task. As shown in Figure 2, the current state is $S_{key}^g$, and the language model has predicted two Person entities as *larry eustachy* and *coach*. At this point, the language model needs to decide whether to continue generating Person entities (the corresponding state is $S_{value}^g$), or stop generating Person entities and further generate other types of entities (the corresponding state is $S_{value}^f$). We convert the current output into natural language as "*In this context, the entity tagged as Person is larry eustachy and coach, there are other r entities*", where $r \in \mathcal{R}^{ent} \cup \varnothing$ represents the type. The natural language has two query forms, $S_s$ and $S_{value}^f$ state is the previous state of $S_{key}^g$, the current key is unknown ($r = \varnothing$), and the query form is "*there are other entities?*". In the $S_{key}^g$ and $S_{value}^g$ states, the current key is known, and $r \in \mathcal{R}^{ent}$ is the label corresponding to the content generated in the subsequent $S_{value}^g$ state, the query form is "*there are other r entities?*".

This conversion process uses natural language to better meet the modeling goals of the language model, summarizing the previous state information and which labels the model should focus on. Then Structure2Text Decoder predicts the probability of the next state based on the context and natural language:

$$\tilde{p}(\mathcal{S}_{i+1}|\mathbf{c}, S_{<i}) = \text{Softmax}(\text{FFN}(\text{Decider}([\mathbf{c}; \text{Convert}(S_{<i})]))), \quad (5)$$

where Decider means Structure2Text Decoder, which is based on the discriminant model. FFN represents the feed-forward network, and Convert represents the Structure2Text operation of converting to natural language.

Finally, combined with the token logits of the language model, the scoring function of the state transition is:

$$\text{Score}(S_{i+1}^j|\mathbf{c}, S_{<i}) = p(\mathbf{y}|\mathbf{c}, S_{<i})[S_{i+1}^j] + \tilde{p}(\mathcal{S}_{i+1}|\mathbf{c}, S_{<i})[S_{i+1}^j]. \quad (6)$$

Based on the process of controllable state transfer, decisions are made according to Equation 4 to perform state transfer during the decoding stage of the language model. This process not only ensures a complete output structure but also aligns information extraction with language model generation.

## 3.5 Iterative Training

During the training phase of the Structure2Text Decoder, we construct training data $\mathcal{D}^g$ based on the ground truth output. As shown in the train data construction in Figure 2, we sample subsets of the extracted results from the ground truth as the state sequence to simulate the state decision process, then convert it into natural language and finally construct the labels of the training data based on whether there is other unextracted content in the context:

$$\hat{p}(d_i^g) = \begin{cases} 1, |\mathcal{E}_i^r| > 0 \\ 0, |\mathcal{E}_i^r| = 0 \end{cases}, \quad (7)$$

where $d_i^g \in \mathcal{D}_i^g$ denotes data constructed from multiple subsets $\mathcal{D}_i^g$ of the ground truth of $d_i$, $r$ denotes the type in $d_i^g$, and $|\mathcal{E}_i^r|$ denotes the number of remaining unextracted content of type $r$. If $r = \varnothing$, $|\mathcal{E}_i^r|$ means the number of remaining unextracted content.

However, in real-world scenarios, the output of the language model is not completely correct. For example, some wrong entities, entity pairs, or arguments may be output in information extraction tasks. The actual decision-making process is more complicated and is missing from the ground truth data. For this reason, we additionally construct training data $\mathcal{D}^p$ based on the predicted output of the language model:

$$\hat{p}(d_i^p) = \begin{cases} 1, |\mathcal{E}_i^r| > 0 \wedge \text{Pre}(d_i^p) \geq \lambda \\ 0, |\mathcal{E}_i^r| = 0 \end{cases}, \quad (8)$$

where $d_i^p \in \mathcal{D}_i^p$ denotes data constructed from multiple subsets $\mathcal{D}_i^p$ of the predicted outputs of $d_i$. $\text{Pre}(d_i^p)$ means the extraction precision of $d_i^p$, and $\lambda$ means the precision threshold.

Finally, we define the training process of the Structure2Text Decoder as multiple rounds of iterative training. As shown in the iterative training decider in Figure 2, we use the output of the ground truth and the prediction results of the language model to construct data to train the Structure2Text Decoder. Then, we use the updated Structure2Text Decoder to help the language model transfer states and construct the next round of training data based on the output. This iterative training process includes more complex correct or error states, which can improve the robustness of the Structure2Text Decoder. The training loss is:

$$\mathcal{L} = \sum_{d_i \in \mathcal{D}} -(\sum_{d_i^g \in \mathcal{D}_i^g} \hat{p}(d_i^g) \log \tilde{p}(d_i^g) + \sum_{d_i^p \in \mathcal{D}_i^p} \hat{p}(d_i^p) \log \tilde{p}(d_i^p)). \quad (9)$$

## 4 Experiment

### 4.1 Experimental Settings

*4.1.1 Datasets.* We focus on NER, RE, and EAE tasks, and conduct experiments on three widely used information extraction datasets: Automatic Content Extraction 2005 (ACE05) [12], Roles Across Multiple Sentences (RAMS) [15], and WikiEvents [28].

ACE05 is a comprehensive dataset derived from a variety of sources including newswires, weblogs, broadcast conversations, and broadcast news. It is commonly utilized for NER, RE, and sentence-level EAE tasks. Specifically, the dataset includes 34,474 manually annotated entities across 7 types, 5,860 entity relations of 7 types, and 5,055 events with 6,040 arguments spanning 33 event types

**Table 2: Experimental results in 0-shot and 20-shot scenarios on ACE05, WikiEvents, and RAMS datasets. The bold text marks the highest value with Llama3.1 8b.**

| Model | Method | NER ACE05 | | | RE ACE05 | | | EAE ACE05 | | EAE WikiEvents | | EAE RAMS | |
|---|---|---|---|---|---|---|---|---|---|---|---|---|---|
| | | P | R | F1 | P | R | F1 | Arg-I | Arg-C | Arg-I | Arg-C | Arg-I | Arg-C |
| | | | | | | | 0-shot | | | | | | |
| GPT-3.5 | Vanilla | 66.59 | 40.97 | 50.73 | 3.35 | 6.92 | 4.52 | 30.51 | 24.26 | 14.35 | 12.25 | 29.19 | 22.72 |
| | Structure | 65.58 | 40.54 | 50.10 | 3.70 | 7.86 | 5.03 | 31.00 | 24.61 | 13.97 | 11.56 | 28.96 | 22.71 |
| GPT-4 | Vanilla | 77.60 | 42.36 | 54.80 | 6.33 | 7.39 | 6.82 | 30.54 | 24.89 | 15.71 | 14.25 | 36.17 | 30.05 |
| | Structure | 76.39 | 42.36 | 54.50 | 6.95 | 8.18 | 7.51 | 31.16 | 25.99 | 15.18 | 13.30 | 36.39 | 30.56 |
| Llama3.1 8b | Vanilla | **54.08** | 28.11 | 36.99 | 5.21 | 1.57 | 2.42 | 26.83 | 18.90 | 14.56 | 12.72 | 23.71 | 17.85 |
| | CoT | 51.33 | 34.47 | 41.25 | 4.99 | 3.14 | 3.86 | 30.20 | 22.45 | 15.13 | 12.38 | 24.25 | 19.13 |
| | Constraint | 51.68 | 32.68 | 40.04 | 4.47 | 7.23 | 5.53 | 31.31 | 22.12 | 15.97 | 12.93 | 21.62 | 16.70 |
| | **STGE (Ours)** | 41.66 | **44.35** | **42.96** | **7.09** | **10.69** | **8.53** | **34.95** | **26.27** | **22.51** | **19.37** | **25.02** | **19.55** |
| | | | | | | | 20-shot | | | | | | |
| GPT-3.5 | Vanilla | 62.89 | 43.19 | 51.21 | 9.68 | 15.57 | 11.93 | 36.17 | 28.29 | 22.68 | 17.94 | 32.98 | 27.30 |
| | Structure | 64.43 | 44.02 | 52.30 | 9.41 | 15.09 | 11.59 | 36.54 | 28.22 | 21.28 | 17.33 | 33.28 | 27.57 |
| GPT-4 | Vanilla | 74.87 | 49.49 | 59.59 | 18.23 | 20.13 | 19.13 | 37.74 | 32.51 | 22.76 | 20.46 | 37.23 | 31.62 |
| | Structure | 74.45 | 49.06 | 59.14 | 18.49 | 21.23 | 19.77 | 37.15 | 31.94 | 23.33 | 20.69 | 36.85 | 30.89 |
| Llama3.1 8b | Vanilla | 70.94 | 45.14 | 55.18 | 17.81 | 15.88 | 16.79 | 41.33 | 32.42 | 25.58 | 21.85 | 33.15 | 27.39 |
| | CoT | 70.44 | 45.41 | 55.22 | 17.31 | 15.41 | 16.31 | 39.01 | 31.21 | 26.22 | 22.67 | 32.94 | 26.95 |
| | Constraint | **75.16** | 47.63 | 58.31 | 18.52 | 16.51 | 17.46 | 41.53 | 32.80 | 25.90 | 22.45 | 33.76 | 28.07 |
| | **STGE (Ours)** | 73.03 | **53.50** | **61.76** | **21.42** | **18.55** | **19.88** | **43.15** | **35.06** | **27.90** | **25.11** | **35.26** | **29.40** |

and 35 argument roles, reflecting its extensive utility for diverse information extraction tasks.

RAMS focuses on document-level EAE tasks, compiled from 12,000 news articles sourced from the Reddit platform. This dataset includes 9,124 events and 21,237 arguments, covering an expansive 139 event types and 65 argument roles, with each event distributed across a context of 5 sentences, offering a unique challenge in document-level event extraction tasks.

WikiEvents, another key dataset for document-level EAE tasks, is constructed from English Wikipedia entries that describe real-world events. It comprises 3,951 events and 5,536 arguments, categorized into 50 event types and 59 argument roles. Each event in this dataset is distributed across the context of the entire document, evaluating the model's ability to extract arguments over long ranges.

For each dataset, we adhere to the official data splits. ACE05 data is processed following the methodology of DyGIE++ [42], whereas RAMS and WikiEvents are pre-processed according to the protocols established by PAIE [37]. In the few-shot scenario, we sample data from the training set as few-shot data.

*4.1.2 Evaluation Metrics.* We employ the same evaluation metrics as previous methods across all tasks. NER task uses Precision (P), Recall (R), and Micro-F1 (F1) metrics for evaluation. An entity is correct if its offsets and type match any entity mention. RE task uses Precision (P), Recall (R), and Micro-F1 (F1) metrics for evaluation. A relation is correct if its entity offsets and relation type match any relation triple. For the EAE task, we use Argument Identification (Arg-I) and Argument Classification (Arg-C) metrics, and use Micro-F1 (F1) for evaluation. An event argument must have correct offsets and event type for Arg-I metric, and additionally the correct role type for Arg-C metrics.

*4.1.3 Implementation Details.* We use three advanced language models: Llama-3.1 (Llama-3.1-8B-Instruct) [14], GPT-3.5 (gpt-3.5-turbo-0125), and GPT-4 (gpt-4-turbo) [1]. We employ Llama-3.1-8B-Instruct as its foundational language model, the max new tokens are set to 256. Our Structure2Text Decider is built based on RoBERTa-large model [31], pre-training on the sampled NERD dataset [10] (a few-shot NER dataset). In the few-shot scenario, Structure2Text Decider is further fine-tuning on the few-shot data and optimized using the AdamW optimizer [34] with learning rates of $2 \times 10^{-5}$. The precision threshold $\lambda$ is set to 0.2 on RAMS and 0.5 otherwise. The Jina embeddings 2 (jina-embeddings-v2-base-en) [17], a mainstream model for long document embeddings, are used to retrieve demonstrations. For fair comparison, all baselines use the same retrieval strategy to retrieve demonstrations, and the number of demonstrations is set to 2. All models and embeddings are accessible via the HuggingFace Transformers library[1] and OpenAI API,[2] respectively. All models are temperature-fixed at 0, utilizing NVIDIA V100 80GB GPUs and PyTorch for implementation.

*4.1.4 Baselines.*

- **Vanilla**: Employing In-context Learning [4] with heuristic rules applied post-processing to filter redundant content, aiming to refine the output structure.
- **CoT**: Employing Chain-of-Thought (CoT) [46] prompts the language model to first think about the labels that exist in the context and then extract the corresponding content.
- **Constraint**: Incorporating Constrained Decoding [19, 28] during the inference phase. It restricts the language model's

---

[1] https://github.com/huggingface/transformers
[2] https://openai.com/api/

**Table 3: Ablation study results in 20-shot scenario on ACE05, WikiEvents, and RAMS datasets.**

| Method | NER ACE05 | | | RE ACE05 | | | EAE ACE05 | | WikiEvents | | RAMS | |
|---|---|---|---|---|---|---|---|---|---|---|---|---|
| | P | R | F1 | P | R | F1 | Arg-I | Arg-C | Arg-I | Arg-C | Arg-I | Arg-C |
| STGE | 73.03 | **53.50** | **61.76** | **21.42** | **18.55** | **19.88** | **43.15** | **35.06** | **27.90** | **25.11** | **35.26** | **29.40** |
| -w/o decider | **75.08** | 47.83 | 58.43 | 20.24 | 16.19 | 17.99 | 42.01 | 33.86 | 26.13 | 22.75 | 34.11 | 28.23 |
| -w/o pre-training | 73.02 | 48.53 | 58.30 | 20.00 | 16.82 | 18.27 | 42.44 | 34.00 | 26.24 | 23.05 | 34.07 | 28.08 |
| -w/o iterative training | 60.76 | 52.77 | 56.48 | 19.80 | 18.24 | 18.99 | 42.20 | 32.53 | 26.64 | 23.89 | 34.87 | 28.70 |

token generation to those from the given context or predefined characters of the output structure.

- **Structure**: Utilizing OpenAI's JSON mode for structured outputs, this approach ensures that all outputs conform to the complete JSON format.

## 4.2 Main Results

Table 2 shows the NER, RE, and EAE experimental results in 0-shot and 20-shot scenarios on ACE05, WikiEvents, and RAMS datasets. We have the following observations and analyses:

**By Aligning Language Model Generation with Structured Information Extraction via Controllable State Transition, our method can significantly outperform the Vanilla baseline.** Our STGE improves F1 by **11.9%~16.1%** over the vanilla baseline in two few-shot settings on the NER task. On the RE task, our STGE even archives an F1 score nearly **three times** higher than the vanilla baseline. Our STGE improves Arg-C F1 by average **32%** over the vanilla baseline in two settings on three EAE datasets. Even when competing against OpenAI's JSON mode (GPT4: Structure), our method also outperforms GPT-4 in most cases, achieving gains of up to 6.07 in Arg-C F1 on the WikiEvents dataset in the 0-shot setting. This verifies that our controllable state transition mechanism can bridge the gap to make the language model better fit structure information extraction.

**While ensuring correct format output, Structure method generally fails to surpass the Vanilla method.** Illustrating that merely adhering to format constraints does not substantially contribute to the performance. Specifically, examining the 0-shot scenario on the ACE05 dataset for the NER task, the Structure method scores an F1 of 50.10 with GPT-3.5 and 54.50 with GPT-4, which are marginally worse or on par with the Vanilla method's scores of 50.73 and 54.80, respectively. This pattern is consistent across different tasks and models, where the gains provided by structural adherence alone are minimal. In contrast, our STGE incorporates controllable state transitions specific to the extraction tasks, maintaining correct formatting and showing a significant performance uplift. For example, in the 20-shot scenario on NER ACE05, our method achieves an F1 of 61.76, outstripping the Vanilla method by a substantial margin.

**Multiple methods exhibit higher precision in the NER task yet demonstrate increased recall in the RE and EAE tasks.** This discrepancy suggests that while the models are precise in identifying named entities, they tend to miss more ground truth entities in NER and generate extraneous or incorrect relations and arguments in the more complex RE and EAE tasks. In contrast, our method, which utilizes a state transition process tailored to

structured text formats, not only enhances the model's ability to accurately judge and extract relevant content but also achieves better overall performance with a more balanced precision and recall across all tasks.

## 4.3 Ablation Study

We conduct ablation studies to investigate the effectiveness of each component, and Table 3 presents the results. Specifically, "w/o decider" removes the Structure2Text Decider and only uses the language model to make state transition decisions, "w/o pre-training" means removing the pre-training process of Decider, and "w/o iterative training" means removing multiple rounds of iterative training and only using the dataset $\mathcal{D}^g$ based on the ground truth output to train the Structure2Text Decider. We have the following analysis:

**(1) Without Decider:** Integrating the Structure2Text Decider proves essential, as its removal leads to a significant decline in performance. For example, the F1 score in the NER task on the ACE05 dataset dropped from 61.76 to 58.43, illustrating a decrease of 3.33 percentage points. This underscores that without the decider, the language model prioritizes textual fluency over accurate structured output, struggling to effectively navigate and extract precise content within the imposed constraints.

**(2) Without Pre-training:** Omitting the pre-training component results in a notable reduction in model efficacy across tasks, with the F1 score in the NER task on ACE05 decreasing from 61.76 to 58.30, a drop of 3.46 percentage points. This highlights the crucial role of pre-training in equipping the model.

**(3) Without Iterative Training:** The absence of iterative training markedly diminishes the model's performance, with the F1 score in the NER task on ACE05 decreasing from 61.76 to 56.48, reflecting a significant reduction of 5.28 percentage points. This component's role is critical in continually adapting and refining the model's responses to complex scenarios, illustrating that iterative adjustments and exposure to error states significantly enhance the decider's decision-making accuracy and robustness.

This provides a clearer view of how each component contributes to the overall effectiveness of the model, clearly demonstrating their necessity in achieving optimal performance in structured information extraction tasks.

## 4.4 Few-shot Results

As shown in Figure 4, we construct experiments and analyses on multiple few-shot NER, RE, and EAE scenarios on ACE05 dataset, where the number of shot $\in [20, 40, 60, 80, 100]$. Specifically, our method achieves the best performance in all scenarios, improving

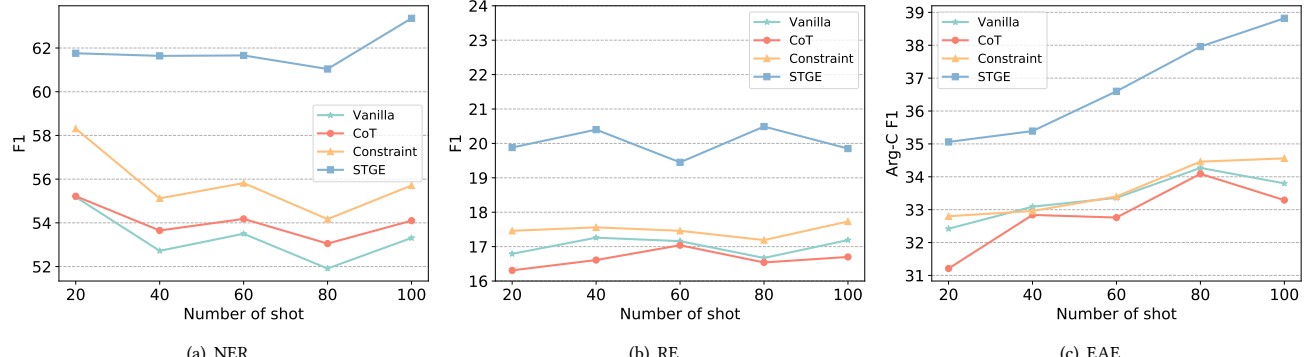

**Figure 4: Experimental results on multiple few-shot NER, RE, and EAE scenarios on ACE05 dataset.**

**Table 4: The ratio of correct structures output by the model.**

| Method | NER ACE05 | RE ACE05 | EAE ACE05 | EAE WikiEvents | EAE RAMS |
|---|---|---|---|---|---|
| w/o rule | 2.10 | 0.00 | 23.82 | 6.85 | 14.81 |
| Vanilla | 80.69 | 52.23 | 84.86 | 47.95 | 67.28 |
| Constraint | 67.91 | 73.00 | 97.77 | 53.15 | 78.42 |
| STGE | **100.00** | **100.00** | **100.00** | **100.00** | **100.00** |

**Table 5: The results of the number of output tokens.**

| Method | NER ACE05 | RE ACE05 | EAE ACE05 | EAE WikiEvents | EAE RAMS |
|---|---|---|---|---|---|
| w/o rule | 241.70 | 253.61 | 127.94 | 188.24 | 210.37 |
| Vanilla | 74.27 | 123.20 | 32.54 | 101.01 | 88.16 |
| Constraint | 82.97 | 83.85 | 34.86 | 111.55 | 61.65 |
| STGE | **37.91** | **45.28** | **20.27** | **17.85** | **18.69** |
| Gold | 28.31 | 23.52 | 10.00 | 11.77 | 18.90 |

the performance by **6.58~10.05** F1, **2.29~3.82** F1, and **2.30~5.02** Arg-C F1 on NER, RE, and EAE tasks. It is worth noting that since the number of shot affects the retrieval space of the demonstration, the baseline methods achieve different degrees of performance fluctuations [38, 45], and even achieve poor results due to hallucinations [21]. In contrast, our method is based on the controllable state transition process. It cleverly incorporates information extraction features into the decision-making process, which significantly alleviates this performance fluctuation and achieves better results, with an average performance improvement of **8.57** F1, **3.00** F1, and **3.38** Arg-C F1 on NER, RE, and EAE tasks. This indicates that Structure2Text Decider can help the language model correct wrong decisions, thereby alleviating hallucinations and preventing the generalization of wrong content.

## 4.5 Format Analysis

We analyze the structure and token length of the model output in the challenging 0-shot scenario. The experimental results are shown in Table 4 and 5. "w/o rule" represents the original output obtained by removing the heuristic rule from the Vanilla baseline, and "Gold" represents the ground truth output.

*4.5.1 Structure Analysis.* In our analysis of model outputs to determine structural accuracy, displayed in Table 4. It's evident that heuristic rules are crucial, without them, as seen in the "w/o rule" method, the model produces excessive irrelevant content. Although constrained decoding, as utilized in the "Constraint" method, narrows the model's search space and somewhat enhances structure accuracy, it fails to consistently deliver optimal results. In stark contrast, our method, employing the controllable state transition

process, consistently achieves **100%** correct structure output across various tasks, demonstrating its superior capability to output precise structure.

*4.5.2 Output Analysis.* Table 5 illustrates the token lengths generated by various models in a 0-shot scenario, revealing that methods without heuristic rules ("w/o rule") and even the "Vanilla" produce excessively verbose outputs, far exceeding efficient token use. In contrast, our method not only minimizes token output but aligns closely with the ground truth ("Gold"), demonstrating its superior efficiency. This efficiency stems from our controllable state transition process, which integrates task-specific features into decision-making, significantly enhancing the model's capability to generate precise and relevant content swiftly, thereby optimizing both accuracy and output conciseness.

## 5 Conclusion

In this paper, we align language model generation with structured information extraction via controllable state transitions. Specifically, we propose controllable state transition process constraints and simplify the language model's generation process. Furthermore, we propose Structure2Text Decider, which uses text that incorporates the features of the information extraction task to help the language model make decisions. Our method not only ensures the correct structured output, but also incorporates the task-specific features, aligning the generation and extraction processes to improve the extraction ability of the model. We conduct extensive experiments on multiple tasks and datasets, and our method achieves superior performance and generation efficiency in multiple scenarios.

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

Received 20 February 2007; revised 12 March 2009; accepted 5 June 2009

