# OpenReview forum: "Bridging the Gap: Aligning Language Model Generation with Structured Information Extraction via Controllable State Transition"
_ACM.org/TheWebConf/2025/Conference — WWW 2025 Oral_

### Official Review · Reviewer_Uc3T · 2024-10-25

**Novelty:** 5
**Technical Quality:** 5

**Review:**

Paper summary: This paper addresses the challenges large language models (LLMs) face in structured information extraction due to differences in task type, output format, and granularity. To bridge this gap, the authors redefine the LLM generation process as controllable state transitions, aligning generation with extraction to maintain output structure. They introduce the Structure2Text decider, which translates structured outputs into natural language, enhances task-specific focus, and reduces errors. Extensive experiments show that this approach significantly improves performance and ensures structured, easily parsed outputs, enhancing data utility on the web.

Pros:
1. This work identifies a notable gap between the information extraction task and the training corpus of large language models (LLMs). The proposed method to bridge this gap is both innovative and engaging.
2. The authors introduce a novel, controllable text generation approach that leverages LLMs alongside a trained decider to determine the task state in information extraction.
3. A comprehensive evaluation experiment is presented to demonstrate the effectiveness of the proposed method, with competitive performance against strong baselines.

Cons:
1. The clarity of certain sections could be improved. For instance, Lines 489–499 contain numerous symbols that make the content challenging to follow.
2. Could the authors elaborate on the absence of a CoT baseline for GPT-4 and GPT -3.5? Additionally, it would be beneficial to understand why the study does not include comparisons with other state-of-the-art NER methods mentioned in the Related Work, even those based on Roberta or BERT backbones.
3. Including a qualitative analysis of outputs before and after implementing the proposed method would help illustrate its impact more clearly.

**Questions:**

Line 122: The authors state that “language models enjoy considerable freedom in text generation, producing content that is open-ended and unstructured.” However, many LLMs, such as CodeLlama and ChatGPT, demonstrate strong capabilities in coding tasks that require formatted and structured text generation. Could the authors elaborate on why they characterize LLMs as primarily unstructured in their content generation?

**Reviewer Confidence:**

3: The reviewer is confident but not certain that the evaluation is correct

**Scope:**

4: The work is relevant to the Web and to the track, and is of broad interest to the community

---

### Official Review · Reviewer_MLU4 · 2024-12-02

**Novelty:** 6
**Technical Quality:** 5

**Review:**

The paper titled "Bridging the Gap: Aligning Language Model Generation with Structured Information Extraction via Controllable State Transition" proposes an innovative framework to bridge the gap between natural language generation (NLG) and structured information extraction (IE). The authors identify three key challenges: task type, output format, and modeling granularity, which they address using a Controllable State Transition model combined with a Structure2Text Decider. The research's goals are clear, and the relevance to practical applications in web data utilization, semantic search, and knowledge base updating is well-argued.

Strengths:

The gap in aligning NLG with IE is a well-motivated issue, and the paper articulates the need for structured outputs for various applications effectively.
The formalization of language model generation as a state transition process is novel and provides a clear mechanism for aligning generative and extractive paradigms.
Weaknesses:

The framing of the task as inherently a transition mechanism is conceptually rich but may oversimplify cases where content generation is not linear or state-dependent.
There is a lack of detailed discussion on how this model performs in scenarios requiring highly nuanced, domain-specific knowledge.

**Questions:**

1. In table 1, why not include CoT, constraint and STGE for the two GPT models as well?

2. What is the difference between structure, CoT and constraint? I do not fully follow the description in the paper.

3. The costs of the iterative approach could be somewhat significant if there are repetitive calls to a large LLM. Is there a way to optimize for the number of state transitions and make it more efficient?

**Reviewer Confidence:**

3: The reviewer is confident but not certain that the evaluation is correct

**Scope:**

4: The work is relevant to the Web and to the track, and is of broad interest to the community

---

### Official Review · Reviewer_1qYB · 2024-12-04

**Novelty:** 6
**Technical Quality:** 5

**Review:**

Paper summary: The paper introduces Structured Transition Guided Extraction (STGE), a novel approach to aligning large language models (LLMs) with structured information extraction tasks. It employs a controllable state transition mechanism to ensure outputs are both linguistically coherent and structurally accurate. The inclusion of the Structure2Text Decider enhances precision and reduces hallucinations by converting structured outputs into natural language. Experiments on Named Entity Recognition (NER), Relation Extraction (RE), and Event Argument Extraction (EAE) demonstrate that STGE outperforms baselines in zero-shot and few-shot settings.

1. Quality: The paper demonstrates strong technical rigor, providing a detailed breakdown of the proposed method and its implementation. The controllable state transition mechanism is innovative, and the experimental results substantiate the method’s effectiveness. However, there are inconsistencies between reported numerical results in the text and corresponding tables, which detract from the reliability of the claims. Additionally, the methodology lacks detailed explanations for heuristic baselines, which hinders reproducibility.

2. Clarity: The paper is generally well-written, with clear descriptions of the technical framework. However, some sections, such as the introduction, are repetitive, and examples in the methodology contain grammatical errors. The lack of explicit researcher names in the related work section diminishes clarity and accessibility. A dedicated discussion of limitations and broader implications is also missing, which would enhance the overall narrative.

3. Originality: The work is highly original in framing structured information extraction as a controllable state transition problem. The Structure2Text Decider is a particularly novel contribution that bridges generative language models with structured tasks effectively. The method distinguishes itself from existing approaches by its focus on ensuring structural accuracy while maintaining linguistic fluency.

4. Significance: The proposed method has substantial implications for structured information extraction, addressing a critical gap in the application of LLMs. The strong experimental performance across multiple datasets and tasks highlights its significance. However, a more thorough discussion of generalization to other domains and scalability to large datasets would strengthen its impact.

List of Pros:
- Innovative Approach: The controllable state transition mechanism is a unique and effective way to align generation with extraction tasks.
- Strong Empirical Results: Demonstrates significant improvements over baselines, including state-of-the-art models like GPT-3.5 and GPT-4.
- Task-Specific Design: The Structure2Text Decider effectively bridges unstructured generation and structured extraction.
- Practical Implications: Aligning LLMs with structured information extraction enhances their usability in real-world applications.

List of Cons:
- Inconsistent Results: Discrepancies between numerical results in the text and tables reduce credibility. Especially, the numerical results in the text appear inconsistent with Table 2. For example, the text claims an F1 improvement of 11.9%~16.1% on the NER task, but Table 2 shows only a 6.58% improvement for Few-Shot (20 Shots). Similar discrepancies are observed for RE (3.09% improvement vs. claimed "nearly three times higher") and Arg-C F1 on EAE datasets (2.63% improvement vs. claimed 32%). Additionally, the bold highlighting of Llama's highest values is misleading, as comparisons include GPT-3.5 and GPT-4 but exclude a direct performance comparison.
- Reproducibility Gaps: Missing details about heuristic rules and Chain-of-Thought prompts for baselines hinder replication.
- Incomplete References: Several cited works ([19], [31], [41], [47]) are outdated and need updates to their published versions.
- Missing Chapters: The paper lacks sections on Discussion, Limitations, and Future Work, which are essential for a comprehensive evaluation.
- Language Issues: Grammatical errors in examples and repetition in the introduction diminish clarity.

Areas for Improvement:
1. Introduction: Avoid repetitions in the contributions section for better engagement.
2. Related Work: Replace generic terms like "researchers" with explicit names to improve clarity.
3. Methodology: Correct grammatical errors in examples and provide heuristic rules for baselines to ensure reproducibility.
4. Experiments: Align reported numerical results with corresponding tables and clarify comparisons with models like GPT-3.5 and GPT-4.

By addressing these issues, the paper would significantly improve in clarity, comprehensiveness, and impact.

**Questions:**

1. How does your methodology compare to simpler rule-based approaches?
2. Are there any constraints or assumptions in the datasets used (e.g., annotation quality) that may affect the generalizability of the method?
3. Why are there discrepancies between the numerical results reported in the text and those shown in the tables?
4. Can you explain the performance gap between STGE and GPT-4 in specific tasks, particularly where GPT-4 outperforms?
5. How does the method handle overfitting in few-shot scenarios, especially when leveraging pre-trained components?
6. What are the heuristic rules applied in the Vanilla baseline, and how might changes to these rules impact the reported performance?
7. How scalable is the method for larger datasets or real-time extraction tasks, and what are its computational requirements?
8. What are the primary limitations of the current approach, and how could they be addressed in future work?
9. How does the proposed method contribute to the broader field of aligning LLMs with structured tasks compared to other state-of-the-art approaches?
10. Do you plan to make the publicly available implementation of STGE and of the pre-trained model for STGE to enable reproduction of the results?

**Reviewer Confidence:**

4: The reviewer is certain that the evaluation is correct and very familiar with the relevant literature

**Scope:**

4: The work is relevant to the Web and to the track, and is of broad interest to the community

---

### Official Review · Reviewer_y77i · 2024-12-06

**Novelty:** 6
**Technical Quality:** 6

**Review:**

This paper presents a novel framework for improving structured information extraction using large language models (LLMs). The authors introduce a Controllable State Transition mechanism and the Structure2Text Decider to align generation and extraction processes. Their method, validated through experiments on NER, RE, and EAE tasks, demonstrates enhanced performance and structural integrity of outputs compared to baseline approaches.

The paper introduces a well-defined framework for aligning generative tasks with structured extraction. The use of a controllable state transition approach is clearly described and supported by a comprehensive methodology. Extensive experiments conducted on multiple datasets (ACE05, RAMS, WikiEvents) provide strong evidence of the proposed method's effectiveness. Metrics are clearly defined and align with the research goals. The contributions are explicitly stated, emphasizing the novelty of the controllable state transition framework and Structure2Text Decider.

The writing is clear and coherent, with well-structured sections detailing the problem, methods, experiments, and results. The paper avoids excessive technical jargon, making it accessible to readers with a foundational understanding of LLMs and information extraction.
Figures and tables are well-designed and provide intuitive visual summaries of the methodology and results. However, the dense formatting in some areas could benefit from additional spacing or reorganization for readability.

The controllable state transition concept and its integration with the Structure2Text Decider sounds like a significant advancement over existing LLM-based extraction techniques. The iterative training strategy for improving robustness adds to the paper's innovative contributions.

**Questions:**

1. How well do you expect the proposed framework to generalize to domains or tasks beyond NER, RE, and EAE? Have you considered testing on less-standard datasets or tasks, such as those in specialized domains (e.g., biomedical or legal text)?
2. Can the proposed state transition mechanism handle tasks with highly dynamic or hierarchical output structures, such as multi-hop reasoning or nested entities?
3. Could you clarify the computational overhead introduced by the iterative training process? How does this compare to other state-of-the-art fine-tuning techniques in terms of efficiency?
4. Why were certain baseline methods (e.g., prompting paradigms other than Chain-of-Thought or fine-tuning approaches) excluded from the comparisons? Would these comparisons yield any insights about your framework's strengths and weaknesses?
5. Beyond structured information extraction, do you foresee applications for the controllable state transition mechanism in other NLP tasks, such as summarization, question answering, or code generation?
6. Does the method you proposed struggle with certain types of ambiguities or inconsistencies in the input data?

**Reviewer Confidence:**

4: The reviewer is certain that the evaluation is correct and very familiar with the relevant literature

**Scope:**

4: The work is relevant to the Web and to the track, and is of broad interest to the community